# Role of HDAC5 Epigenetics in Chronic Craniofacial Neuropathic Pain

**DOI:** 10.3390/ijms25136889

**Published:** 2024-06-23

**Authors:** Sifong Elise Hui, Karin N. Westlund

**Affiliations:** Department of Anesthesiology & Critical Care Medicine, University of New Mexico Health Sciences Center, Albuquerque, NM 87131, USA

**Keywords:** trigeminal, neural regeneration, nerve injury, miRNA, cold allodynia

## Abstract

The information provided from the papers reviewed here about the role of epigenetics in chronic craniofacial neuropathic pain is critically important because epigenetic dysregulation during the development and maintenance of chronic neuropathic pain is not yet well characterized, particularly for craniofacial pain. We have noted that gene expression changes reported vary depending on the nerve injury model and the reported sample collection time point. At a truly chronic timepoint of 10 weeks in our model of chronic neuropathic pain, functional groupings of genes examined include those potentially contributing to anti-inflammation, nerve repair/regeneration, and nociception. Genes altered after treatment with the epigenetic modulator LMK235 are discussed. All of these differentials are key in working toward the development of diagnosis-targeted therapeutics and likely for the timing of when the treatment is provided. The emphasis on the relevance of time post-injury is reiterated here.

## 1. Introduction

Epigenetic modifications involving histone deacetylases (HDACs) play a critical role in the molecular mechanisms involved in the transition and establishment of unremitting neuropathic pain [1]. Histones are alkaline proteins that bind DNA without affecting gene sequence. Direct histone modification via acetylation, methylation, ubiquitination, and phosphorylation governs long-term gene expression [2]. The acetylation and deacetylation of histones are integral to gene transcription because they remodel the chromatin to permit greater access to specific DNA regions, thereby controlling gene expression, cell cycle, and signal transduction [3]. Histone acetyltransferases (HATs) and HDACs interactions must be continuously balanced so that homeostasis is preserved [4]. HATs promote chromosomal transcription by unraveling DNA from histones and allowing greater access to the genes. Conversely, HDACs restrain chromosomal transcription by compacting DNA around histones and limiting gene access. Nervous tissue possesses HDACs, and expression of HDACs increases in injured dorsal root ganglia (DRG), in spinal cords exposed to noxious formalin or complete Freund’s adjuvant (CFA), and after spinal nerve ligation [5,6,7,8,9,10].

Among the four classes of HDAC modulators, the two most extensively studied are the zinc-dependent ubiquitous Class I (HDAC 1–3 and 8) and the Class II (HDAC 4–7, 9, and 10) HDACs, whose inhibition reduces pain. Class I (SAHA, MS-275), Class II (SAHA, valproic acid), and pan-(trichostatin A, vorinostat) HDAC inhibitors (HDACi) are effective as pre-treatment for pain. Class III (sirtuins) HDACs are protective against pain, and inhibiting them increases pain. Overexpressing SIRT3 in the spinal dorsal horn prior to chronic constriction injury (CCI) neuropathic pain model induction effectively reduces CCI-instigated mechanical and thermal hypersensitivity in the initial 2 weeks, while SIRT3-/- mice demonstrated elevated pain-related measures [11]. In that same study, the SIRT3 protein, which decreased with nerve injury, was increased in the spinal cord by electroacupuncture administered every other day for 2 weeks. In two previous studies, we have noted that the global RNA profile varies depending on the sample collection time point. Analysis at a chronic 10-week timepoint in our model of chronic craniofacial neuropathic pain provides functional groupings of genes varying from shorter timepoints at 3 and 21 days published previously. Genes altered after treatment with epigenetic modulator LMK235, including those potentially contributing to anti-inflammation, nerve repair/regeneration, and nociception, are discussed.

## 2. Effect of LMK235 Post-Treatment on Behavioral Hypersensitivity, TG Electrophysiological Response, and Molecular Profile in a Chronic Trigeminal Neuropathic Pain Model

Our studies of trigeminal nerve injury [1,12] revealed numerous expressed genes, including ones distinctive to trigeminal ganglia (TG) compared to those reported for inflammatory CFA and sciatic nerve models [13,14,15,16]. While HDAC Class I inhibitors are not effective given as post-treatments, the Class IIa HDAC4/5 inhibitor LMK235 was highly effective in achieving long-lasting reversal of chronic trigeminal neuropathic pain when given as a post-treatment [12]. Numerous other alterations noted in our study following LMK235 treatment may have contributed, including the decreased activation of TG neurons in vitro and the molecular changes. Provisional patent is filed as US-22002211--00334400226655-A1

## 3. Effects of Class IIa HDACi LMK235 on Craniofacial Nerve Injury Hypersensitivity

Promotion of analgesia is among the numerous actions reported for HDAC inhibitors [17]. The in vivo and in vitro effects of Class IIa HDACi LMK235 reported by us recently [12] were examined during the clinically relevant chronic phase at 3 weeks post-injury in two orofacial trigeminal neuropathic pain mouse models utilizing two strains of mice (C57Bl/6, BALBc/cAnNHsd). LMK235 post-treatment at 3 weeks attenuated craniofacial mechanical and cold hypersensitivity for at least the next 7 ensuing weeks in the FRICT-ION model. We established that subcutaneously administered LMK235 is not only effective in alleviating mechanical allodynia but also in preventing the development of anxiety- and depression-like behaviors associated with the chronic trigeminal neuropathic pain models in untreated mice. LMK235 was less efficacious and somewhat slower in returning distortions toward the naïve baseline in female mice, which is congruent with known sex differences in craniofacial pain [18,19,20,21]. Thus, LMK235 or a related drug may ease some of the major clinical effects endured by patients who have chronic trigeminal neuropathic pain in the future.

The HDAC4/5 inhibitor LMK235 (5 or 10 mg/kg; 7 daily doses in week 3) reversed chronic neuropathic pain behaviors to naïve baseline within 2 weeks in male mice and near baseline in female mice in our study [12]. The durable reversal persisted for >7 weeks of testing in our chronic trigeminal nerve injury FRICT-ION (foramen rotundum inflammatory compression trigeminal infraorbital nerve) model [12] which is particularly easy to induce [22]. Non-evoked chronic pain-related anxiety behaviors that typically emerge after 6 weeks of ongoing pain fail to develop in the LMK235-treated mice. These net results were reflected in reduced neuronal excitability in TG primary cultures treated with LMK235 compared to untreated TG neurons.

## 4. Electrophysiological Responses of TG to LMK235

Examination of the mechanism of action of LMK235 using electrophysiological analyses established that LMK235 reduced the excitability of trigeminal neurons. A shift with ~20% of the small neurons becoming classified as high threshold (rheobase > 200 pA) under LMK235-treated conditions was observed with the in vitro characterization profile, while neurons under control conditions did not have a high threshold. Small-diameter, high-threshold neurons are thought to be nociceptors. The neurons were isolated from mice 3 weeks after FRICT-ION injury and exposed in vitro for 1 h to the HDACi LMK235 prior to these recording sessions.

## 5. Molecular Profile Alterations with Trigeminal Nerve Injury and LMK235 Treatment

Detailed gene profiling of TG after LMK235 treatment of mice with FRICT-ION chronic nerve injury was strikingly similar to that of naïve mice in week 10, in contrast to that of untreated FRICT-ION mice with chronic neuropathic pain [12]. The RNAseq and Western blot data in our trigeminal nerve injury study found 2-fold increased expression of Hdac5 RNA and protein [12]. The significant increase in Hdac5 RNA and protein was not found in the mice treated with the selective Class IIa HDAC4/5 inhibitor LMK235. While LMK235 inhibits HDAC4 in other studies, no Hdac4 was evident in TG RNA profiles for any of our group screens at 10 weeks. Thus, while it is known that nerve injury enhances direct modification of histones through HDACs ability to inhibit gene transcription, the actions of HDACi LMK235 effectively diminished post-injury increases in HDAC5 protein and Hdac5 RNA, as well as promoted or inhibited most other gene alterations in our study. The findings indicate HDAC5 is a critical component in neuronal activation and pain-related behavior in our chronic trigeminal nerve injury neuropathic pain model. This is in contrast to the increased HDAC2 that was reported to be prominent in DRG in the spinal nerve ligation model (SNL) at 3 weeks [10].

## 6. Corroboration of the Molecular Profile with Behavioral and TG Electrophysiological Responses

Reversal of behavioral hypersensitivity threshold and trigeminal neuron excitability with Class IIa HDACi LMK235, as well as the increased Hdac5 RNA and protein levels in nerve-injured TG, provide strong data corroborating HDAC5 epigenetic regulation of craniofacial neuropathic pain. Importantly, the molecular profiles examined during the more clinically relevant chronic pain phase at 10 weeks post-nerve injury indicate the ability of LMK235 to diminish cytokines and increase neuronal repair mechanisms that contribute to chronic neuropathic pain. Several pain-related genes (P2rx4, Cckbr, growth hormone (Gh), and schlafen (Slfn4)) upregulated in the trigeminal nerve injury FRICT-ION model are diminished to naïve or below naïve levels by the LMK235 by week 7 [12].

### 6.1. KCNE2

Ion channel gene *Kcne2* was upregulated 11.10-fold (*p* = 1.24 × 10^−2^) in FRICT-ION mice compared to naïve and greatly but insignificantly reduced (1.56-fold, *p* = 6.10 × 10^−1^) in LMK235-treated mice compared to untreated FRICT-ION mice [12]. In a study with chronic oxaliplatin exposure over 15 days [23], rats developed hypersensitivity with upregulated MiRP1 (KCNE2) expression in DRGs correlating to greater HCN1 channel activity, conductance, and I_h_.

### 6.2. P2X4R and P2X7R

Glial crosstalk with neurons transpires when the injury-induced neuroactivator ATP is released by satellite glia and neurons in the TG [24] and keratinocytes peripherally [25]. ATP signaling in nociception has been shown to involve macrophages [24,26,27] and be functionally coupled to P2X7 [27] and P2X4 receptors [25,26,28]. P2X4 receptors are a critical target expressed in plenitude on C and A-fiber sensory neurons [25,28]. Expression of P2X4R (purinergic receptor P2X4) in the brain, spinal cord, and nerves is associated with pain and anxiety [26,29,30,31]. Upregulation of *P2rx4* RNA at 10 weeks after trigeminal nerve compression injury (2.12-fold, *p* = 1.28 × 10^−2^) was not observed in mice treated with LMK235 3 weeks after injury [12]. Contrarily, LMK235 downregulated *P2rx4* RNA expression. Thus decrease in P2X4R would be equivalent to the effects of P2X4R antagonists, which have been shown to enhance spatial memory, subdue microglial activation, and decrease cytokine release in a traumatic brain injury model when administered continuously for three consecutive days starting at the time of model induction [32]. Macrophage and Schwann cell P2X4R also promote cytokine and brain-derived neurotrophic factor release implicated in injured nerve regeneration and remyelination [26,33]. P2X7 receptors, primarily found on immune cells and microglia [34,35], are upregulated [34,36,37] and have a significant role in chronic inflammatory and neuropathic pain [34,35,37]. Notably, tolerance to morphine generated through repetitive dosing is accompanied by an upregulation of P2X7R [34,35] such that P2X7R inhibition reestablishes morphine analgesia [35]. Similar to P2X4R, Schwann cell P2X7R promotes axon regrowth and myelination [35]. Surprisingly, there were only minimal and insignificant fold-changes for *P2rx7* in LMK235-treated and untreated FRICT-ION mice in our recent study [12] possibly due to the utilization of the BALBc/cAnNHsd strain of mice. Numerous studies with C57Bl/6 mice in migraine and trigeminal neuralgia mouse models have revealed that P2X7R is an important regulator of pain. One report by Kushnir et al. found that P2X7R expression in the TG of BALB/c mice was not increased one week post-CFA-induced submandibular orofacial inflammation [38], concurring with the *P2rx7* findings in our study [12]. Chen et al. demonstrated that blocking the cortical P2X7R channel alone is not sufficient for abolishing spreading depolarization in a rodent migraine model involving BALB/c mice [39]. It is the P2X7-PANX1 pore complex that is vital for modulating spreading depression.

### 6.3. CCKBR

Cholecystokinin B receptor (CCKBR) and cholecystokinin (CCK) in sensory ganglia, spinal cord, and supraspinal neurons mediate nociception, morphine insensitivity, and anxiety in numerous rodent pain models [40,41,42,43,44,45]. CCK was elevated in DRG two weeks post-sciatic nerve transection and was correlated with morphine insensitivity [41]. *Cckbr* gene expression was upregulated in mice TG 3 days (4.00-fold, *p* = 1.03 × 10^−4^) and 21 days (2.72-fold, *p* = 9.17 × 10^−4^) following TIC trigeminal nerve injury compared to naïve mice in our previous study [1]. Morphine analgesia was reinstated, and opioid tolerance was overturned with CCKBR antagonists [40,44]. CCKBR has been demonstrated in the literature to play an essential role in anxiety and panic disorder [45,46].

### 6.4. Somatostatin

Somatostatin possesses anti-nociceptive properties [47,48] and is extensively present in a myriad of tissue types, including DRG [48]. Genes for somatostatin and its receptors were not drastically altered in our recent LMK235 study for either untreated or treated FRICT-ION mice [12] suggesting chronic pain pathophysiology in TG may differ from DRG. Hypersensitivity prevailing in a clinically relevant chronic neuropathic pain model for 6–10 weeks has not been mitigated in prior studies utilizing HDAC inhibitors as post-treatments. Post-treatment with Class IIa HDACi LMK235 at 3 weeks post-trigeminal nerve injury effectively reversed hypersensitivity, and its effectiveness persisted long-term over two months of study. The HDACi post-treatment at 8 weeks is the first evidence of diminished mechanical hypersensitivity at such a long-term time point after chronic neuropathic pain. These data suggest that further additional recovery may have been accomplished by continuing the HDACi LMK235 treatment.

HDAC4 and 5 have previously been implicated in rodent models of inflammatory, neuropathic, and thermal pain [49,50,51,52,53]. LMK235, used intrathecally at day 7 post-spinal ligation in a previous study in rats, was shown to reduce hypersensitivity with equal effectiveness but was tested for only 5 h [53]. In that study with rats, as in our studies with mice, treatment with LMK235 did not change the levels of HDAC4 itself after spinal ligation. However, a variety of other HDAC4 alterations are described in that spinal nerve ligation study in rats, including the prevention of a 14-3-3β-mediated cytoplasmic shift of phosphorylated HDAC4 from the nucleus and from astrocytes to neurons in the ipsilateral spinal cord [53]. In another study, both male and female HDAC4 KO mice had greater response latencies in the hot plate assay [49].

Initially developed as cancer therapeutics [3,4,54], some HDACi compounds have been tested for the treatment of neurodegenerative diseases and as anti-inflammatory agents [55,56,57,58]. Four HDAC inhibitors are presently FDA-approved to treat some forms of cancer [59]. To date, a number of studies have implicated other epigenetic changes in RNA expression associated with the development of acute hypersensitivity in rodent pain models (reviewed in [16]). Numerous previous reports have noted that Class I HDAC inhibitors (SAHA, MS-125) pre-treatments reverse von Frey thresholds induced by nerve injury. Class I and II HDACi pre-emptively delivered intrathecally prior to footpad CFA or nerve injury prevented inflammatory thermal hypersensitivity [9,60]. The H3K9 methylation inhibitor UNC0642 prevents H3K9 methylation and has been demonstrated to significantly alleviate visceral hyperalgesia in a rat model [61]. Class I HDACi MS-275 increased acetylated H3K9 [9,60] in the spinal dorsal horn but failed to alter hypersensitivity if administered after the injury [60], while in another study HDAC inhibitors were found to be effective for inflammatory hypersensitivity [62]. Pre-treatment with pan-HDAC I/II inhibitors reduced the second-phase formalin test response by upregulating the inhibitory glutamate mGlu2 receptor in DRG [7], as well as abrogating nerve injury and inflammatory hypersensitivity when given prior to or simultaneously with pain model induction [9,60,63]. It was also noted that Class II HDACi given intrathecally reduced heat hypersensitivity on the footpad in mice with CFA inflammation [9]. However, Zhang et al. noted that epigenetic mechanisms are operational in chronic but not acute pain models [8].

While Class I HDACi have only been shown to attenuate pain-related behavior and mechanical hypersensitivity with pre-treatment, the Class IIa HDACi LMK235 was highly effective long-term as a post-treatment. Spinal knockdown of HDAC5 has been shown previously to alleviate long-term hypersensitivity in a sciatic nerve injury neuropathic pain model [51], reducing pain-related behavior in a spared nerve injury model [64], whereas pain-like behaviors are observed in naïve mice with HDAC5 spinal overexpression [51]. Epigenetic silencing of K+ channels in DRG neurons followed spinal nerve ligation in another report showing histone acetylation decrease and methylation increase at Kcna4, Kcnd2, Kcnq2, and Kcnma1 gene promoters [10]. In that study by Laumet et al., inhibition of G9a methyltransferase reactivated 40 of 42 silenced genes associated with K+ channels while normalizing 638 genes down- or up-regulated by nerve injury. HDAC4 and 5 are linked to pain-induced stress behavior and addiction in rodent models [64,65]. HDAC4 is reportedly a critical mediator of inflammation-associated thermal hypersensitivity [49,50]. Together, previous findings indicate significant epigenetic modulation is occurring during the transition from acute to chronic pain, and our studies [1,12] demonstrate differential epigenetic modulation in chronic trigeminal neuropathic pain models at three different time points (3 days, 3 weeks, and 7 weeks).

## 7. Effects on Anxiety- and Depression-Like Behaviors

Anxiety-like behaviors did not develop in our studies [12] with mice tested in weeks 6–8 after the LMK235 treatment was given daily in week 3. This indicated that the absence of mechanical and cold hypersensitivity provided by LMK235 prevented the development of anxiety-like behavior. SNI-induced anxiety- and depression-like phenotypes are also diminished in mice when Hdac5 is genetically deleted [64].

Chronic craniofacial pain is strongly associated with anxiety and depression [66,67]. HDACi are used in psychiatry and neurology as mood stabilizers and anti-epileptics [68,69]. Opioids, antidepressants, and anticonvulsants (carbamazepine, α2δ-1 Ca2+ channel ligand gabapentin, pregabalin) either alone or in combination offer therapeutic benefit for many neuropathic pain syndromes [70,71]. In a major proportion of patients with trigeminal neuropathic pain, however, these drugs and even third-line opioids are ineffective or patients become refractory to them, leading patients to submit to surgical approaches or radiofrequency nerve ablations that provide relief that is not permanent in most cases [67,72,73]. It has been suggested that repeated doses of much less specific HDACi’s (trichostatin A, valproic acid) could potentially reestablish systemic morphine analgesia and be useful as adjuvant analgesic treatment when combined with morphine [74]. No addictive potential was found for LMK235 in our recent study [12].

## 8. Epigenetic Dysregulation and Genes Differentially Regulated in Chronic Craniofacial Pain

Debilitating craniofacial neuropathic pain is prevalent in 7–15% of the population [21,75] and the incidence is reportedly double in females compared to males [20,21,76]. Craniofacial pain can result from facial blunt force trauma, endodontics, oral surgery, infections/inflammation, or unknown causes [73,76,77]. Some patients with continuous craniofacial neuropathic pain report a history of trauma such as traffic accidents or dental procedures like tooth extractions [78]. Sixty-seven percent of patients with preoperative neuropathic pain develop trigeminal neuropathic pain following microsurgical intervention to repair a nerve injury [79]. As with other chronic neuropathic pain conditions, there is typically no evidence of any persisting tissue pathology for trigeminal neuralgia [80]. Patients with craniofacial neuropathic pain often visit multiple doctors and undergo many tests without establishing an accurate diagnosis, even when the condition is disabling [67,77,80]. More frustrating to the patient is that multiple treatments, such as periodontal surgery, root canal therapy, and even multiple tooth extractions, typically provide no relief or even increase pain [67,77,81].

The abundance of HDACs in the central nervous system and the involvement of epigenetic regulation in pain are well documented [5,6,7,10,82]. As an important regulator of gene transcription, H3K9 turns on genes when acetylated and turns off genes when methylated [2,10]. Global deacetylation of H3K9 takes place at lysine residue 9 of histone 3, catalyzed by HDACs [82]. Histone deacetylation causes chromatin compaction and decreased transcriptional activity in neuropathic pain models [2,5,7,10,82]. One measure of epigenetic regulation of HDACs in neuropathic pain is the alteration in immunophenotypic global H3K9 acetylation (H3K9ac) expression in NeuN dual-labeled TG neurons, satellite glia, microglia, oligodendrocytes, and activated astrocytes post-nerve injury in studies with mice or rats [1,63,82,83]. The relative abundance of H3K9ac was elevated at 30 min in mice intrathecally injected with SAHA or MS-275 [9]. The quantities of Class IIa HDACs 4,5,7, and 9 were raised in the spinal cord immunoblot analysis subsequent to CFA-induced hindpaw thermal hypersensitivity [9]. Comparably, decreased H3K9 acetylation was observed 3 weeks after trigeminal nerve compression injury in mice in our previous study [1], further indicating the HAT/HDAC balance was shifted during acute and chronic timepoints. In the rat study by Wei et al. [82], H3K9ac was also significantly decreased in TG in the TREZ trigeminal nerve compression model of craniofacial neuropathic pain in week 2. The diminished TG H3K9ac was replenished in TG treated with Class I/II HDACi carbamazepine in that study. Together, these studies provide evidence that epigenetic dysregulation of gene H3K9 expression by Class I and II HDACs is related to the long-term persistence of pain.

## 9. How RNA Alterations Are Associated with Improvements in Trigeminal Nerve In-Jury Models

Table 1 provides detailed information for mouse genes up- and down-regulated that are also reported in human neuralgia. Group comparisons are shown for the FRICT-ION trigeminal neuropathic pain versus (vs.) naïve mice in the column with BLACK font, the FRICT-ION mice treated with LMK235 vs. naive mice in RED font, and the LMK235-treated vs. untreated FRICT-ION mice in GREEN font.

The most prominent pain-related genes altered in the FRICT-ION mice with mechanical hypersensitivity included significant decreases in inhibitory pain neurotransmitter serotonin-related RNA for serotonin transporter (*Slc6a4*; −4.86-fold, *p* = 4.23 × 10^−6^) and serotonin synthetic enzyme tryptophan hydroxylase 2 (*Tph2*; −4.51-fold, *p* = 1.27 × 10^−7^). The *Slc6a4* was upregulated by an equal amount in the LMK235-treated mice [12]. This clearly implies a decrease in serotonin function and aligns with FRICT-ION-induced hypersensitivity. Slc6a4 is similarly reported to affect a patient’s propensity for developing trigeminal neuralgia (TN), intensity of pain, and response to carbamazepine treatment [84].

*Cacna1a*, encoding a calcium channel, mediates trigeminal synaptic transmission by regulating neurotransmitter release, and a gain-of-function in this gene was correlated with TN in a patient [85]. Di Stefano et al. uncovered a Cacna1a variant in a patient with TN [86]. Corresponding with this, *Cacna1a* was upregulated 1.10-fold (*p* = 6.89 × 10^−2^) in FRICT-ION mice compared to naïve mice, while it was downregulated 1.16-fold (*p* = 3.24 × 10^−3^) in LMK235-treated FRICT-ION mice compared to untreated FRICT-ION mice [12].

Li et al. demonstrated that the following four microRNAs, miR-132-3p, miR-146b-5p, miR-155-5p, and miR-384, were elevated in the serum of patients with TN [87]. Genes associated with neuropathic hypersensitivity were among those regulated by the four microRNAs and included Ndnf, Ncald, Nfasc, Nrp2, Nrep, Nacc2, Net1, Nrg2, Nova1, Nras, Nf2, Ngef, Nrg3, Nav3, Nts, Nkain2, Neurod4, and Nrf2. None of the four microRNAs, miR-132-3p2, miR-146b-5p3, miR-155-5p4, and miR-384, were present in our LMK235 mice data [12]. The three microRNA genes found in our recent mouse study were Mir22hg, Mir6236, and Mir703. Neuropathic pain-related genes upregulated in LMK235-treated FRICT-ION mice compared to untreated FRICT-ION mice were *Net1* (1.12-fold, *p* = 1.68 × 10^−1^), *Nrg2* (1.10-fold, *p* = 4.16 × 10^−1^), *Nova1* (1.04-fold, *p* = 3.01 × 10^−1^), *Nf2* (1.04-fold, *p* = 2.22 × 10^−1^), and *Neurod4* (1.57-fold, *p* = 1.47 × 10^−1^). Downregulated neuropathic pain related genes in LMK235-treated FRICT-ION mice compared to untreated FRICT-ION mice were *Nras* (1.03-fold, *p* = 5.21 × 10^−1^), *Ngef* (1.34-fold, *p* = 7.33 × 10^−7^), *Nrg3* (1.00-fold, *p* = 9.77 × 10^−1^), *Nav3* (1.19-fold, *p* = 2.68 × 10^−2^), *Nts* (1.74-fold, *p* = 4.23 × 10^−7^), and *Nkain2* (1.14-fold, *p* = 5.34 × 10^−2^).

Whole-exome sequencing of patients revealed that genetic mutations causing perturbations in the structure or function of ion channels [86,88,89] and signaling in the GABAergic system [88] correlated with TN [86,88,89]. Ion channel genes upregulated in LMK235-treated FRICT-ION mice compared to untreated FRICT-ION mice were *Scn8a* (1.05-fold, *p* = 1.98 × 10^−1^), *Kcnk1* (1.05-fold, *p* = 2.82 × 10^−1^), *Scn9a* (1.29-fold, *p* = 2.62 × 10^−4^), *Trpc6* (1.08-fold, *p* = 6.17 × 10^−1^), *Trpm3* (1.19-fold, *p* = 2.47 × 10^−2^), *Trpm4* (1.05-fold, *p* = 6.82 × 10^−1^), and *Clcn2* (1.01-fold, *p* = 8.92 × 10^−1^) [12]. Strikingly, a few of these results did not align with previous studies demonstrating that an Scn8a variant with gain-of-function may be instrumental in TN development [89] and decreased Kcnk1 expression was associated with reduced hypersensitivity [90]. RNA sequencing was performed using the spinal cord and DRG in the study by Eigenbrod et al. [90]. Consistent with our recent findings [12], ipsilateral gingiva in TN patients had decreased expression of Nav1.7 (encoded by *Scn9a*) [91]. Ion channel genes downregulated in LMK235-treated FRICTION mice compared to untreated FRICT-ION mice were *Cacna1h* (1.49-fold, *p* = 2.21 × 10^−5^), *Scn2a/Scn2a1* (1.04-fold, *p* = 3.85 × 10^−1^), *Scn3a* (1.01-fold, *p* = 9.15 × 10^−1^), *Scn7a* (1.07-fold, *p* = 6.29 × 10^−1^), *Trpv6* (1.19-fold, *p* = 3.74 × 10^−1^), *Kcna5* (1.41-fold, *p* = 1.33 × 10^−2^), *Kcnd2* (1.00-fold, *p* = 9.66 × 10^−1^), *Kcnh7* (1.19-fold, *p* = 2.36 × 10^−1^), *Kcnj6* (1.38-fold, *p* = 2.75 × 10^−2^), *Kcns2* (1.25-fold, *p* = 5.91 × 10^−2^), *Cacna1d* (1.04-fold, *p* = 5.84 × 10^−1^), *Cacna1g* (1.43-fold, *p* = 3.12 × 10^−4^), *Cacna1i* (1.39-fold, *p* = 4.03 × 10^−6^), and *Cacnb1* (1.13-fold, *p* = 1.13 × 10^−2^). Analogously, Kang et al. found that spared nerve injury in rats upregulated *Cacna1h*-encoded CaV3.2 T-type calcium channels in the DRG, provoking allodynia [92]. There was also increased expression of Nav1.3 (encoded by *Scn3a*) in the gingiva of patients with TN [91].

It is imperative to note that there are genetic mutations that are not associated with neuropathic pain. Although *Ntrk1*, encoding a nerve growth factor receptor, was upregulated 1.80-fold (*p* = 9.90 × 10^−4^) in LMK235-treated FRICT-ION mice compared to untreated FRICT-ION mice [12], the NTRK1/rs633 polymorphism was not specifically connected to TN [93]. SCN9A/rs6746030 polymorphism was also not more prominent in TN patients compared to control patients [93].

Charcot–Marie-Tooth (CMT) is a neurological disorder that is not typically associated with cranial neuropathy. However, 37% of CMT patients from a familial cohort afflicted with TN harbored the Mpz mutation G163R [94]. Mpz was downregulated 1.05-fold (*p* = 9.47 × 10^−1^) in LMK235-treated FRICT-ION mice compared to untreated FRICT-ION mice [12].

Trigeminal pain-related evoked responses differed between healthy subjects with low-activity monoamine oxidase type A (MAOA) and high-activity MAOA [95]. Participants genotyped with low-activity MAOA had distorted serotoninergic pain modulation manifested in aberrant neuronal activation, excitability, and habituation [95]. *Maoa* was upregulated 1.14-fold (*p* = 2.64 × 10^−3^) in LMK235-treated FRICT-ION mice compared to untreated FRICT-ION mice [12].

Peripheral neuroectomy in patients with TN revealed that bone morphogenetic proteins (BMPs) are expressed in the trigeminal nerve [96]. mRNA for Bmp2, Bmp3, Bmp4, and Bmp5 were all present in the myelin sheaths of Schwann cells, while only *Bmp2* mRNA was found in the nerve fibers of trigeminal nerves [96]. Earlier publications proposed that BMPs might regulate neurogenesis [97], peripheral nerve regeneration [98], and peripheral nerve homeostasis maintenance [96]. *Bmp2*, *Bmp3*, and *Bmp5* were all downregulated 1.02- (*p* = 8.75 × 10^−1^), 2.70- (*p* = 7.00 × 10^−5^), and 1.01-fold (*p* = 9.64 × 10^−1^) in LMK235-treated FRICT-ION mice compared to untreated FRICT-ION mice, respectively [12].

Naturally, some genes corresponding with human TN did not appear or have differences in expression foldchanges in our LMK235 mouse study [12].

The GO and heatmap analyses in our recent study [12] provide insight into potential mechanisms for LMK235’s ability to reverse the behavioral and neuronal recording indicators of chronic neuropathic pain. Prominent are wound healing genes, transcription factors including numerous sequence-specific DNA binding terms (Hoxc8, Hoxb9, Hoxd8, Hoxd9, Hoxa6, Hoxb8, Hoxb7, Hoxc6, Hoxb6), neuronal/myelin repair, ion channel, and immune/cytokine genes.

## 10. Anti-Inflammation

HDACi have been shown to reduce the production of the cytokines IL-1β, IL-6, IFNγ, and TNF-α [99,100]. One source of cytokines in the TG is the M2 macrophages, which secrete IL-10 and TGF-β, promoting healing and repair for neurons and Schwann cells [101]. M2 macrophages invade the TG/DRG after nerve injury and proliferate around ATF3-expressing neurons in weeks 1–2 [102,103,104]. Satellite glia also proliferate in injured TG/DRG and express IL-1, CCL-2, and TNF-α [102,105,106,107].

## 11. Axonal Repair

The impact of macrophages, satellite glia, and certain cytokines on neuronal damage and repair is well known [101,108,109]. Peripheral nerves possess self-repair capabilities, but those with marked damage or substantial defects are challenging to repair extrinsically [110,111]. Central nervous system nerves have no or limited repair capacity [112,113]. Investigating the pathophysiology of peripheral nerve repair is important for the clinical treatment of peripheral nerve restoration and regeneration.

The RNA profiles and GO analyses for our recent study indicate axonal repair is a factor in nerve recovery following injury [12]. The remyelination-associated gene *Sostdc1* is strikingly upregulated by Class IIa histone deacetylase and known to play important roles in synaptic plasticity, axon regeneration, neurite extension, and neural differentiation. Sostdc1 is highly expressed in the developing optic fiber layer, optic nerve, and ganglion cells of the human eye [114] and its gene expression was proposed as a biomarker for rheumatoid arthritis [115] but our recent study is likely to be the first report of such an increase after chronic trigeminal nerve injury [12]. HDAC5 plays a role in the regulation of gene transcription involved in inhibition of neurite elongation, cellular/system adaptations to chronic emotional stimuli, and drug-induced circuitry changes [64,65,116,117,118]. Nuclear export of HDAC5 by the back-propagated calcium influx after nerve injury is followed by its transport to damaged nerve endings, where it has been found to be essential for nerve regeneration and neuronal plasticity [119,120]. HDAC5-deacetylated tubulin in microtubules following peripheral sciatic nerve ligation promotes growth cone formation for axon regeneration [121]. In contrast, tubulin deacetylation did not occur in ligated optic nerves or DRG from mice with hemisected spinal cords, demonstrating neuronal regeneration via injury-induced tubulin deacetylation is limited to the peripheral nervous system [121]. In another model of nerve injury, model-specific Hdac6 genetic deletion from sensory neurons was not shown to avert cisplatin-induced mechanical hypersensitivity, while global knockout of HDAC6 was protective [122,123]. This was interpreted to signify a role of HDAC6 in other cell types since depletion of MRC1 (CD206)-positive macrophages locally decreased the ability of an HDAC6 inhibitor to reverse cisplatin-induced mechanical allodynia [123]. While microglia were not affected, M2-macrophage-dependent spinal cord Il-10 mRNA and signaling were increased with the HDAC6 inhibitor.

NMDA receptor-interacting protein α2δ-1 (encoded by the *Cacna2d1a* gene), the mechanism of action targeted by gabapentin and pregabalin, is increased in the DRG of rats with L5 and L6 nerve ligation and enhances spinal cord synaptic NMDA receptor activity, resulting in hypersensitivity [70]. Other data provided in that Zhang et al. study found long-lasting hypersensitivity evident with HDAC2 knockdown is reversible with gabapentin, signifying HDAC2 is a transcriptional repressor of neuropathic pain [70]. We have shown previously that H3K9 acetylation decreases in the TG of nerve-injured mice [1]. The increase in early intermediate gene *Atf3* in the gene profile [1] predicts its assistance with induction of regeneration-associated gene (RAG) response genes that promote peripheral nerve regeneration and neurite outgrowth via Vip, Ngf, Grp, Gal, and Pacap governed by Atf3 [124]. Microarray validation in our previous study identified two RAGs, Sprr1a and Gal, differentially regulated in the TG after induction of TIC craniofacial neuropathic pain [1]. Only some of the genes in the Gey et al. study [124] were found in our recent RNAseq data set (*Vip*, *Gal*, and *Grp*) after acute trigeminal neuropathy [12].

Facial nerve regeneration was diminished in Atf3 mutant mice, and primary dorsal root ganglia neurons that were ATF3-deficient had less neurite outgrowth [124]. ATF3 was already upregulated 7.84-fold at 3 days after TIC trigeminal nerve injury in our previous study [1].

## 12. Other Molecular and Epigenetic Mechanisms Implicated

Puttagunta and colleagues have detailed specific epigenetic changes promoting axonal regeneration [125]. Subsequent to a peripheral but not a central axonal injury, histone acetyltransferase p300/CBP-associated factor (PCAF) was found to elicit histone 3 lysine 9 acetylation at the promoters of known essential regeneration-associated genes. PCAF-dependent regenerative gene reprogramming necessitates retrograde signaling in the extracellular signal-regulated kinase (ERK) pathway. In that study, GAP-43, galanin, and BDNF gene expression were increased in L4-L6 DRG at days 1, 3, and 7 post-sciatic nerve axotomy in quantitative ChIP assays. Increased gene expression was found for H3K9ac and PCAF, while H3K9Me2 gene expression was decreased at the promoters for GAP-43, galanin, and BDNF at day 1 post-sciatic nerve axotomy. GAP-43, galanin, and BDNF gene expression increases were correlated with PCAF increases at the promoters of these activated regeneration genes. Significant upregulation of genes for galanin (*Gal*; 25.96-fold, 2.40-fold) and ATF3 (*Atf3*; 7.84-fold, 2.74-fold) was reported in our gene profile for the TIC trigeminal nerve injury model at day 3 and day 21 post-injury, respectively [1]. Decreased H3K9ac was observed by day 21 in TIC mice [1]. These same genes had minimal and insignificant foldchanges in week 10 in the RNA profile for mice with untreated chronic FRICT-ION neuropathic pain (Gal, 1.10-fold, *p* = 4.27 × 10^−1^; Atf3, 1.09-fold, *p* = 7.26 × 10^−1^; Bdnf, −1.02-fold, *p* = 8.02 × 10^−1^; Gap−43, −1.03-fold, *p* = 4.92 × 10^−1^) [12]. Additional key mechanisms for the efficacy of Class IIa histone deacetylases have been investigated in another study, including the 14-3-3β-mediated cytoplasmic shift of phosphorylated HDAC4 from the nucleus and from astrocytes to neurons in the ipsilateral spinal cord [53].

## 13. Pituitary Hormone with Nociceptive Actions

Growth hormone (*Gh*) RNA was increased in the TG of FRICT-ION mice (8.06-fold, *p* = 4.46 × 10^−3^) compared to naïve mice but greatly decreased in LMK235-treated mice (25.22-fold, *p* = 9.19 × 10^−5^) compared to untreated FRICT-ION mice [12]. Previously, Gh RNA expression was reported to be present specifically in TG but not DRG in male C57BL/6 mice [126]. In that same report, masseter muscle or dural administration of exogenous growth hormone (GH) evoked 4 days of craniofacial mechanical allodynia in a dose-dependent manner that was alleviated by a growth hormone receptor antagonist, pegvisomant, used clinically. Exogenous GH reportedly possesses anti-nociceptive properties in the spinal/DRG system [126,127]. Our findings support those who propose that the trigeminal system has different nociceptive signaling pathways as well as pathophysiological outcomes versus those for spinal nociception [12,126]. Other pituitary hormone RNAs with nociceptive actions, including prolactin, have been reported numerous times, but only in female mice [128,129].

## 14. Epigenetic Balance

The degree of histone acetylation is dictated by the balance between HAT and HDAC activity (Figure 1). The importance of timing post-injury in pain-related studies has not been properly considered. An exception was presented in the discussion for a study of sodium channel Nav1.6 expression in a type 2 diabetes model with comparisons to acute nerve injury studies [130]. Nerve injury models produce acute and abrupt alterations of many relevant features, typically including decreased content of many pain-related molecules such as Nav1.6. In the Ren study, Nav1.6 was examined at post-natal 5 months in diabetic mice. The percentage of Nav1.6-positive cells (62.9 ± 5.5%) was significantly higher at this chronic stage of innate type 2 diabetes compared to controls and younger mice [130]. The importance of epigenetic balance was emphasized in a study with HDAC5 mutant mice in which HDAC5 knockout resulted in heightened cocaine reward behavior and increases in steady-state mRNA levels of specific genes [131]. The study concluded that loss of HDAC5 provided vulnerability to the transition from short-term to long-term hypersensitivity effects of repeated cocaine. Furthermore, chronic stress with a social defeat paradigm and cocaine addiction downregulated HDAC5 in the nucleus accumbens, while acute exposures to stress and cocaine did not. The relevance of the acetylation/deacetylation balance was also accentuated in a study of chronic morphine administration [132].

Figure 1 is a pictorial diagram displaying HDAC5 associated events that occur following nerve injury. Gene transcription is modified via histone deacetylation when nerve injury occurs, such that various genes promoting the conversion from acute to chronic pain are upregulated.

The type of information provided in the papers reviewed here is critically important because epigenetic dysregulation during the development and maintenance of chronic neuropathic pain is not yet well characterized, particularly for craniofacial pain. We have noted that gene expression changes reported vary depending on the nerve injury model and the reported sample collection time point. Future studies should provide clear timeline information to inform readers more precisely about the context of the reported changes. These dependent differentials are key in working toward the development of diagnosis-targeted therapeutics and perhaps the timing of the treatment as a variable.

## 15. Conclusions

Our studies [1,12] address the knowledge gap about how epigenetic modifications in TG neurons contribute to chronic neuropathic pain. Our findings further emphasize the relevance of Class IIa HDAC5 epigenetic modification in the maintenance of chronic neuropathic craniofacial pain [12]. Both of our studies [1,12] and findings from others present convincing data that Class I, II, and Class III HDACs are pertinent for preventing and treating acute and chronic neuropathic pain. In our study inhibiting the HDAC5 epigenetic response with Class IIa HDACi, the LMK235 post-treatment reduced neuronal activation as well as both evoked and non-evoked pain-related behaviors [12]. The efficacy of Class IIa HDACi post-treatment was demonstrated both during the transition from acute to chronic pain in week 3 and after long-term chronic pain for 8 weeks in two rodent models. LMK235 is efficacious because it halts deacetylation while decreasing HDAC5 and other pain-related gene expressions. Class IIa HDACs 4 and 5 are unique because they travel between the nucleus and the cytoplasm, while other HDACs are ordinarily confined to either the nucleus or the cytoplasm [4,120]. While pre-treatment effectiveness is reported for Class I, II, and pan-HDACi in many studies [7,9,60,63,133], the Class IIa HDACi post-treatment reversal of mechanical and cold hypersensitivity suggests restoration of the histone acetylase/deacetylase balance. The reversal of these key characteristics in our studies, reminiscent of clinical trigeminal neuropathic pain [1,12], indicates the potential therapeutic relevance of moderated use of Class IIa HDAC inhibitors for the alleviation of chronic neuropathic pain.

## Figures and Tables

**Figure 1 ijms-25-06889-f001:**
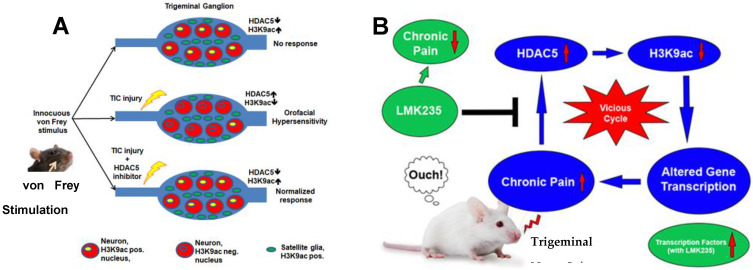
Depiction of the epigenetic response of HDAC5 altering gene transcription that supports the persistence of chronic trigeminal pain and the reversal of chronic neuropathic pain with LMK235 *post*-treatment. (**A**) In naïve mice, the application of innocuous von Frey stimulation (at arrow) does not elicit a response. After TIC trigeminal infraorbital nerve compression injury, hypersensitive response to mechanical stimulation on the face in the form of head withdrawal is observed accompanied by an increase in HDAC5. (**B**) If HDAC5 inhibitor LMK235 is given daily in week 3 during the transition from acute to chronic pain to reduce HDAC5, hypersensitivity on the face is reversed to baseline level. This figure was generated by the authors to provide context.

**Table 1 ijms-25-06889-t001:** Mouse genes that are also reported in human trigeminal neuralgia are shown in the Table below. Shading indicates decreased expression, while no shading indicates increased expression. Yellow highlighted values have >1.75-fold change.

Mouse Genes Up- and Down-Regulated Also Reported in Human Trigeminal Neuralgia		FRICT-ION Vs. Naïve			LMK235-Treated FRICT-ION Vs. Na ï ve			LMK235-Treated FRICT-ION Vs. Untreated FRICT-ION		
*Gene* *Symbol*	NAME	Fold-Change	2-Value	FDR	Fold-Change	* p * -Value	FDR	Fold-Change	* p * -Value	FDR
*Arfgef2*	ADP-ribosylation factor guanine nucleotide-exchange factor 2 (brefeldin A-inhibited)	1.01	8.02 × 10^−1^	9.38 × 10^−1^	1.02	6.82 × 10^−1^	9.30 × 10^−1^	1.01	8.74 × 10^−1^	9.65 × 10^−1^
*Astn2*	astrotactin 2	−1.02	7.72 × 10^−1^	9.26 × 10^−1^	1.04	5.61 × 10^−1^	8.87 × 10^−1^	1.06	3.80 × 10^−1^	7.47 × 10^−1^
*Bmp2*	bone morphogenetic protein 2	1.13	3.27 × 10^−1^	6.66 × 10^−1^	1.11	4.04 × 10^−1^	8.21 × 10^−1^	−1.02	8.75 × 10^−1^	9.65 × 10^−1^
* Bmp3 *	bone morphogenetic protein 3	2.20	1.57 × 10^−3^	2.81 × 10^−2^	−1.23	4.34 × 10^−1^	8.35 × 10^−1^	−2.70	7.00 × 10^−5^	4.66 × 10^−3^
*Bmp4*	bone morphogenetic protein 4	1.13	5.55 × 10^−1^	8.28 × 10^−1^	1.18	4.15 × 10^−1^	8.27 × 10^−1^	1.05	8.23 × 10^−1^	9.50 × 10^−1^
*Bmp5*	bone morphogenetic protein 5	1.87	2.02 × 10^−2^	1.54 × 10^−1^	1.85	2.16 × 10^−2^	4.22 × 10^−1^	−1.01	9.64 × 10^−1^	9.89 × 10^−1^
*Cacna1a*	calcium channel; voltage-dependent; P/Q type; alpha 1A subunit	1.10	6.89 × 10^−2^	3.15 × 10^−1^	−1.06	2.67 × 10^−1^	7.47 × 10^−1^	−1.16	3.24 × 10^−3^	6.64 × 10^−2^
*Cacna1d*	calcium channel; voltage-dependent; L type; alpha 1D subunit	1.01	8.68 × 10^−1^	9.59 × 10^−1^	−1.03	7.06 × 10^−1^	9.36 × 10^−1^	−1.04	5.84 × 10^−1^	8.58 × 10^−1^
*Cacna1g*	calcium channel; voltage-dependent; T type; alpha 1G subunit	1.42	3.73 × 10^−4^	9.63 × 10^−3^	−1.00	9.71 × 10^−1^	9.95 × 10^−1^	−1.43	3.12 × 10^−4^	1.38 × 10^−2^
*Cacna1h*	calcium channel; voltage-dependent; T type; alpha 1H subunit	1.34	1.91 × 10^−3^	3.17 × 10^−2^	−1.11	2.69 × 10^−1^	7.48 × 10^−1^	−1.49	2.21 × 10^−5^	2.07 × 10^−3^
*Cacna1i*	calcium channel; voltage-dependent; alpha 1I subunit	1.21	6.55 × 10^−3^	7.32 × 10^−2^	−1.14	6.20 × 10^−2^	5.50 × 10^−1^	−1.39	4.03 × 10^−6^	6.44 × 10^−4^
*Cacnb1*	calcium channel; voltage-dependent; beta 1 subunit	1.11	3.79 × 10^−2^	2.24 × 10^−1^	−1.02	6.74 × 10^−1^	9.28 × 10^−1^	−1.13	1.13 × 10^−2^	1.41 × 10^−1^
*Clcn1*	chloride channel 1	−1.27	2.53 × 10^−1^	5.99 × 10^−1^	−1.33	1.67 × 10^−1^	6.68 × 10^−1^	−1.05	8.20 × 10^−1^	9.49 × 10^−1^
*Clcn2*	chloride channel 2	−1.12	8.61 × 10^−2^	3.54 × 10^−1^	−1.11	1.11 × 10^−1^	6.26 × 10^−1^	1.01	8.92 × 10^−1^	9.69 × 10^−1^
*Clic5*	chloride intracellular channel 5	−1.03	7.54 × 10^−1^	9.18 × 10^−1^	−1.08	4.43 × 10^−1^	8.40 × 10^−1^	−1.05	6.50 × 10^−1^	8.86 × 10^−1^
*Eef2*	eukaryotic translation elongation factor 2	−1.06	3.53 × 10^−1^	6.89 × 10^−1^	−1.05	4.62 × 10^−1^	8.50 × 10^−1^	1.01	8.47 × 10^−1^	9.58 × 10^−1^
*Gabra5*	gamma-aminobutyric acid (GABA) A receptor; subunit alpha 5	1.26	1.75 × 10^−4^	5.70 × 10^−3^	1.02	7.71 × 10^−1^	9.53 × 10^−1^	−1.23	4.81 × 10^−4^	1.88 × 10^−2^
* Gabra6 *	gamma-aminobutyric acid (GABA) A receptor; subunit alpha 6	−1.77	2.14 × 10^−1^	5.55 × 10^−1^	1.18	7.11 × 10^−1^	9.38 × 10^−1^	2.08	1.08 × 10^−1^	4.53 × 10^−1^
*Gabre*	gamma-aminobutyric acid (GABA) A receptor; subunit epsilon	−1.33	1.16 × 10^−1^	4.12 × 10^−1^	−1.22	2.73 × 10^−1^	7.50 × 10^−1^	1.10	6.13 × 10^−1^	8.71 × 10^−1^
*Gabrg1*	gamma-aminobutyric acid (GABA) A receptor; subunit gamma 1	1.11	2.41 × 10^−1^	5.86 × 10^−1^	1.26	7.39 × 10^−3^	3.41 × 10^−1^	1.14	1.31 × 10^−1^	4.93 × 10^−1^
*Jakmip1*	janus kinase and microtubule interacting protein 1	1.12	3.13 × 10^−2^	2.00 × 10^−1^	1.02	7.51 × 10^−1^	9.47 × 10^−1^	−1.10	6.08 × 10^−2^	3.44 × 10^−1^
*Kcna5*	potassium voltage-gated channel; shaker-related subfamily; member 5	1.33	4.39 × 10^−2^	2.42 × 10^−1^	−1.06	6.70 × 10^−1^	9.27 × 10^−1^	−1.41	1.33 × 10^−2^	1.56 × 10^−1^
*Kcnc3*	potassium voltage gated channel; Shaw-related subfamily; member 3	−1.01	7.82 × 10^−1^	9.31 × 10^−1^	−1.05	3.14 × 10^−1^	7.75 × 10^−1^	−1.04	4.65 × 10^−1^	8.01 × 10^−1^
*Kcnd2*	potassium voltage-gated channel; Shal-related family; member 2	1.09	8.36 × 10^−2^	3.47 × 10^−1^	1.09	9.02 × 10^−2^	5.93 × 10^−1^	−1.00	9.66 × 10^−1^	9.90 × 10^−1^
*Kcnh2*	potassium voltage-gated channel; subfamily H (eag-related); member 2	−1.01	7.51 × 10^−1^	9.17 × 10^−1^	−1.06	1.81 × 10^−1^	6.81 × 10^−1^	−1.04	3.06 × 10^−1^	6.85 × 10^−1^
*Kcnh7*	potassium voltage-gated channel; subfamily H (eag-related); member 7	1.23	1.87 × 10^−1^	5.21 × 10^−1^	1.03	8.65 × 10^−1^	9.73 × 10^−1^	−1.19	2.36 × 10^−1^	6.29 × 10^−1^
*Kcnj6*	potassium inwardly-rectifying channel; subfamily J; member 6	1.26	1.19 × 10^−1^	4.18 × 10^−1^	−1.10	5.37 × 10^−1^	8.80 × 10^−1^	−1.38	2.75 × 10^−2^	2.33 × 10^−1^
*Kcnk1*	potassium channel; subfamily K; member 1	−1.07	1.67 × 10^−1^	4.93 × 10^−1^	−1.02	7.50 × 10^−1^	9.47 × 10^−1^	1.05	2.82 × 10^−1^	6.67 × 10^−1^
*Kcns2*	K+ voltage-gated channel; subfamily S; 2	1.40	5.66 × 10^−3^	6.63 × 10^−2^	1.12	3.62 × 10^−1^	7.99 × 10^−1^	−1.25	5.91 × 10^−2^	3.40 × 10^−1^
*Kcnv1*	potassium channel; subfamily V; member 1	1.16	4.56 × 10^−1^	7.64 × 10^−1^	1.26	2.23 × 10^−1^	7.14 × 10^−1^	1.09	6.37 × 10^−1^	8.81 × 10^−1^
*Kif1b*	kinesin family member 1B	−1.00	9.89 × 10^−1^	9.97 × 10^−1^	1.01	8.81 × 10^−1^	9.76 × 10^−1^	1.01	8.69 × 10^−1^	9.64 × 10^−1^
*Maoa*	monoamine oxidase A	−1.06	1.96 × 10^−1^	5.33 × 10^−1^	1.08	9.03 × 10^−2^	5.93 × 10^−1^	1.14	2.64 × 10^−3^	5.76 × 10^−2^
*Mapk3*	mitogen-activated protein kinase 3	−1.02	6.05 × 10^−1^	8.52 × 10^−1^	−1.02	5.92 × 10^−1^	9.00 × 10^−1^	−1.00	9.87 × 10^−1^	9.96 × 10^−1^
*Mpz*	myelin protein zero	1.13	8.73 × 10^−1^	9.61 × 10^−1^	1.07	9.25 × 10^−1^	9.85 × 10^−1^	−1.05	9.47 × 10^−1^	9.85 × 10^−1^
*Nacc2*	nucleus accumbens associated 2; BEN and BTB (POZ) domain containing	−1.05	2.73 × 10^−1^	6.19 × 10^−1^	−1.05	2.68 × 10^−1^	7.48 × 10^−1^	−1.00	9.92 × 10^−1^	9.97 × 10^−1^
*Nav3*	neuron navigator 3	1.33	5.07 × 10^−4^	1.21 × 10^−2^	1.12	1.87 × 10^−1^	6.85 × 10^−1^	−1.19	2.68 × 10^−2^	2.29 × 10^−1^
*Ncald*	neurocalcin delta	1.01	6.84 × 10^−1^	8.89 × 10^−1^	1.05	1.74 × 10^−1^	6.75 × 10^−1^	1.03	3.41 × 10^−1^	7.16 × 10^−1^
*Ndnf*	neuron-derived neurotrophic factor	1.24	2.98 × 10^−2^	1.95 × 10^−1^	1.31	5.67 × 10^−3^	3.29 × 10^−1^	1.06	5.55 × 10^−1^	8.48 × 10^−1^
*Net1*	neuroepithelial cell transforming gene 1	−1.24	9.15 × 10^−3^	9.20 × 10^−2^	−1.11	2.12 × 10^−1^	7.07 × 10^−1^	1.12	1.68 × 10^−1^	5.46 × 10^−1^
*Neurod4*	neurogenic differentiation 4	−1.78	6.79 × 10^−2^	3.13 × 10^−1^	−1.13	6.76 × 10^−1^	9.29 × 10^−1^	1.57	1.47 × 10^−1^	5.16 × 10^−1^
*Nf2*	neurofibromatosis 2	−1.06	5.82 × 10^−2^	2.87 × 10^−1^	−1.02	4.86 × 10^−1^	8.60 × 10^−1^	1.04	2.22 × 10^−1^	6.12 × 10^−1^
*Nfasc*	neurofascin	−1.07	1.40 × 10^−1^	4.55 × 10^−1^	−1.09	6.15 × 10^−2^	5.50 × 10^−1^	−1.02	6.94 × 10^−1^	9.06 × 10^−1^
*Ngef*	neuronal guanine nucleotide exchange factor	1.26	9.49 × 10^−5^	3.66 × 10^−3^	−1.06	3.13 × 10^−1^	7.75 × 10^−1^	−1.34	7.33 × 10^−7^	1.68 × 10^−4^
*Nkain2*	Na+/K+ transporting ATPase interacting 2	1.20	8.88 × 10^−3^	9.05 × 10^−2^	1.05	4.84 × 10^−1^	8.59 × 10^−1^	−1.14	5.34 × 10^−2^	3.22 × 10^−1^
*Nova1*	neuro-oncological ventral antigen 1	−1.00	9.95 × 10^−1^	9.99 × 10^−1^	1.04	3.05 × 10^−1^	7.68 × 10^−1^	1.04	3.01 × 10^−1^	6.81 × 10^−1^
*Nras*	neuroblastoma ras oncogene	1.03	5.55 × 10^−1^	8.28 × 10^−1^	−1.00	9.66 × 10^−1^	9.94 × 10^−1^	−1.03	5.21 × 10^−1^	8.31 × 10^−1^
*Nrep*	neuronal regeneration related protein	1.03	4.81 × 10^−1^	7.83 × 10^−1^	1.04	2.79 × 10^−1^	7.54 × 10^−1^	1.01	7.07 × 10^−1^	9.11 × 10^−1^
*Nrg2*	neuregulin 2	−1.06	6.54 × 10^−1^	8.74 × 10^−1^	1.04	7.24 × 10^−1^	9.41 × 10^−1^	1.10	4.16 × 10^−1^	7.71 × 10^−1^
*Nrg3*	neuregulin 3	1.09	1.55 × 10^−1^	4.76 × 10^−1^	1.09	1.60 × 10^−1^	6.66 × 10^−1^	−1.00	9.77 × 10^−1^	9.93 × 10^−1^
*Nrp2*	neuropilin 2	1.00	9.64 × 10^−1^	9.90 × 10^−1^	1.07	3.21 × 10^−1^	7.79 × 10^−1^	1.07	3.41 × 10^−1^	7.16 × 10^−1^
* Ntrk1 *	neurotrophic tyrosine kinase; receptor; type 1	−1.95	1.89 × 10^−4^	6.02 × 10^−3^	−1.09	6.17 × 10^−1^	9.10 × 10^−1^	1.80	9.90 × 10^−4^	3.13 × 10^−2^
* Nts *	neurotensin	2.05	2.09 × 10^−10^	1.29 × 10^−7^	1.18	1.60 × 10^−1^	6.66 × 10^−1^	−1.74	4.23 × 10^−7^	1.16 × 10^−4^
*Plcl1*	phospholipase C-like 1	−1.00	9.54 × 10^−1^	9.87 × 10^−1^	1.12	4.57 × 10^−2^	5.13 × 10^−1^	1.13	3.94 × 10^−2^	2.78 × 10^−1^
*Scn2a1*	sodium channel; voltage-gated; type II; alpha 1	1.15	2.10 × 10^−3^	3.37 × 10^−2^	1.10	2.62 × 10^−2^	4.41 × 10^−1^	−1.04	3.85 × 10^−1^	7.51 × 10^−1^
*Scn3a*	sodium channel; voltage-gated; type III; alpha	1.12	1.51 × 10^−1^	4.71 × 10^−1^	1.11	1.82 × 10^−1^	6.82 × 10^−1^	−1.01	9.15 × 10^−1^	9.76 × 10^−1^
*Scn5a*	sodium channel; voltage-gated; type V; alpha	1.23	2.78 × 10^−1^	6.24 × 10^−1^	1.23	2.59 × 10^−1^	7.41 × 10^−1^	1.01	9.76 × 10^−1^	9.93 × 10^−1^
*Scn7a*	sodium channel; voltage-gated; type VII; alpha	1.15	3.32 × 10^−1^	6.71 × 10^−1^	1.07	6.19 × 10^−1^	9.10 × 10^−1^	−1.07	6.29 × 10^−1^	8.78 × 10^−1^
*Scn8a*	sodium channel; voltage-gated; type VIII; alpha	−1.12	2.49 × 10^−3^	3.80 × 10^−2^	−1.07	8.10 × 10^−2^	5.76 × 10^−1^	1.05	1.98 × 10^−1^	5.85 × 10^−1^
*Scn9a*	sodium channel; voltage-gated; type IX; alpha	−1.28	7.02 × 10^−4^	1.54 × 10^−2^	1.01	8.54 × 10^−1^	9.73 × 10^−1^	1.29	2.62 × 10^−4^	1.21 × 10^−2^
* Slc6a4 *	solute carrier family 6 (neurotransmitter transporter; serotonin); member 4	−4.86	4.23 × 10^−6^	3.07 × 10^−4^	−1.11	7.48 × 10^−1^	9.47 × 10^−1^	4.38	1.56 × 10^−5^	1.55 × 10^−3^
*Trak1*	trafficking protein; kinesin binding 1	−1.04	2.17 × 10^−1^	5.57 × 10^−1^	−1.05	1.11 × 10^−1^	6.26 × 10^−1^	−1.01	7.24 × 10^−1^	9.16 × 10^−1^
*Trpc6*	transient receptor potential cation channel; subfamily C; member 6	−1.17	3.27 × 10^−1^	6.66 × 10^−1^	−1.08	6.16 × 10^−1^	9.10 × 10^−1^	1.08	6.17 × 10^−1^	8.73 × 10^−1^
*Trpm2*	transient receptor potential cation channel; subfamily M; member 2	−1.08	1.00 × 10^−1^	3.83 × 10^−1^	−1.10	3.87 × 10^−2^	4.85 × 10^−1^	−1.02	6.78 × 10^−1^	9.00 × 10^−1^
*Trpm3*	transient receptor potential cation channel; subfamily M; member 3	−1.07	4.08 × 10^−1^	7.33 × 10^−1^	1.12	1.57 × 10^−1^	6.66 × 10^−1^	1.19	2.47 × 10^−1^	2.19 × 10^−1^
*Trpm4*	transient receptor potential cation channel; subfamily M; member 4	−1.19	1.12 × 10^−1^	4.06 × 10^−1^	−1.14	2.32 × 10^−1^	7.18 × 10^−1^	1.05	6.82 × 10^−1^	9.02 × 10^−1^
*Trpm7*	transient receptor potential cation channel; subfamily M; member 7	1.04	3.95 × 10^−1^	7.22 × 10^−1^	1.10	5.15 × 10^−2^	5.27 × 10^−1^	1.06	2.72 × 10^−1^	6.59 × 10^−1^
*Trps1*	trichorhinophalangeal syndrome I (human)	1.11	1.13 × 10^−1^	4.06 × 10^−1^	1.14	3.61 × 10^−2^	4.81 × 10^−1^	1.03	6.17 × 10^−1^	8.73 × 10^−1^
* Trpv4 *	transient receptor potential cation channel; subfamily V; member 4	1.76	3.27 × 10^−1^	6.66 × 10^−1^	2.12	1.91 × 10^−1^	6.87 × 10^−1^	1.21	7.37 × 10^−1^	9.21 × 10^−1^
*Trpv6*	transient receptor potential cation channel; subfamily V; member 6	1.10	6.11 × 10^−1^	8.54 × 10^−1^	−1.07	7.19 × 10^−1^	9.40 × 10^−1^	−1.19	3.74 × 10^−1^	7.42 × 10^−1^
*Unc80*	unc−80 homolog (C. elegans)	−1.03	5.86 × 10^−1^	8.44 × 10^−1^	1.07	1.84 × 10^−1^	6.84 × 10^−1^	1.10	6.07 × 10^−2^	3.44 × 10^−1^

No shading indicates increased expression. Shading indicates decreased expression. Highlighted have >1.75 fold-change.

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
