# Peer review of "Role of HDAC5 Epigenetics in Chronic Craniofacial Neuropathic Pain"

_ijms, 2024, doi:10.3390/ijms25136889_

Round 1

Reviewer 1 Report

Comments and Suggestions for Authors

Dear Authors,

There are just some minor remarks for this really nice manuscript from my side:

1) include the Plan at the beginning of the manuscript;

2) indicate please for the Fig. 1 is it developed by the authors or the sourse from what this Fig. is adopted;

3) Conclusions. Shorten them please. You can also move the last pararaph for the end of Discussion.

4) there are 3 previous century references (out of 133). Well, dont fit very nicely for this otherwise nowadays manuscript. Could you please to remove them or exchange them with the nowadays ones?

Author Response

  • include the Plan at the beginning of the manuscript;
    1. Now provided at the end of the Introduction

2) indicate please for the Fig. 1 is it developed by the authors or the sourse from what this Fig. is adopted;

            This figure was generated by the authors to provide context.  This note now appears at the end of the figure legend.

3) Conclusions. Shorten them please. You can also move the last pararaph for the end of Discussion.  The last paragraph has been moved to the end of the Discussion as suggested, and the Conclusion has been shortened.

4) there are 3 previous century references (out of 133). Well, dont fit very nicely for this otherwise nowadays manuscript. Could you please to remove them or exchange them with the nowadays ones?

Significant and relevant research did not begin in 2000, although the reviewer may be unaware of significant science.

Reviewer 2 Report

Comments and Suggestions for Authors

In this review  the authors assessed the role epigenetic in chronic facial pain 

Introduction : the purpose of the study is clear in the abstract but not in the introduction section which need to be explained and clarified .

Table 1 is too complex , needs more explanation and  should be better alleviated 

conclusion : please be more precise as the reader may be lost between a general effect or only the effect og TG nerve 

Please describe the axis for future research in this area especially in chronic craniofacial nerve pain 

Comments on the Quality of English Language

No issue with english 

Author Response

Reviewer 2

See the italicized comments below addressing the changes and/or additions to the review:

Introduction : the purpose of the study is clear in the abstract but not in the introduction section which need to be explained and clarified .

Added to the end of the discussion:       In two previous studies we have noted the global RNA profile varies dependent on the sample collection time point. Analysis at a chronic 10 week timepoint in our model of chronic craniofacial neuropathic pain provides functional groupings of genes varied from shorter timepoints at 3 and 21 days published previously. Genes altered after treatment with epigenetic modulator LMK235, including those potentially contributing to anti-inflammation, nerve repair/regeneration, and nociception, are discussed.

 Table 1 is too complex , needs more explanation and  should be better alleviated

            Explanatory notations for the Table are now also provided as text in the paragraph just before the Table to clarify:

Table 1 provides detailed information for mouse genes up- and down-regulated that are also reported in human neuralgia. Group comparisons are shown for the FRICT-ION trigeminal neuropathic pain versus (vs) naïve in the column with black font, FRICT-ION mice treated with LMK235 vs naive mice in RED font, and the LMK235 treated vs untreated FRICT-ION mice in GREEN font.“

 Also the text below has been added as the Table Legend:

Table 1. Mouse genes that are also reported in human trigeminal neuralgia are shown in the Table below. Shading indicates decreased expression, while no shading indicates increased expression. Yellow highlighted values have >1.75-fold change.

  Conclusion : please be more precise as the reader may be lost between a general effect or only the effect og TG nerve

Shortening the Conclusion as suggested by Reviewer 1 provides more clarity, as does noting only trigeminal nerve injury in the Conclusion.

Please describe the axis for future research in this area especially in chronic craniofacial nerve pain

Moving the last paragraph of the Conclusion as the end of the Discussion and the addition of the following sentence addresses this concern.